# Comparison of Tinnitus Handicap Inventory and Tinnitus Functional Index as Treatment Outcomes

Marta Fernández [1], María Cuesta [2] , Ricardo Sanz [1] and Pedro Cobo [2,*]

1  Department of Medicine, Faculty of Biomedical and Health Sciences, Universidad Europea de Madrid, 28670 Madrid, Spain
2  Institute of Physical and Information Technologies (ITEFI), CSIC, Serrano 144, 28006 Madrid, Spain
*  Correspondence: pedro.cobo@csic.es; Tel.: +34-915618806

**Abstract:** Background: Tinnitus is an audiological disorder for which there are no objective measuring tools. Thus, many self-report questionnaires have been proposed to assess its severity. These questionnaires have been judged for their capacity to assess the tinnitus severity at baseline, their sensitivity to treatment-related changes (responsiveness), and their resolution. Methods: The most widely used questionnaires for clinical and research studies are the Tinnitus Handicap Inventory (THI) and the Tinnitus Functional Index (TFI). While both questionnaires have been recognized as good evaluators of the baseline tinnitus severity, the latter is considered to be more responsive to changes following treatments. Objectives: The aim of this work is to provide a preliminary comparison of the performance of both questionnaires in the initial and final tinnitus severity assessment of a cohort of patients undergoing a four-month Enriched Acoustic Environment (EAE) therapy. Results: The EAE therapy provided a 30 and 26 point reduction in THI and TFI, respectively. A good correlation is obtained between the THI and TFI questionnaires at baseline and after the treatment. Conclusion: At baseline, the THI provided a higher score than the TFI for a higher degree of tinnitus but a lower score for lower tinnitus severity. Both THI and TFI were good questionnaires for baseline assessment and for treatment-related changes. The THI provided a slightly higher score drop than the TFI following the treatment, although the TFI had better resolution.

**Keywords:** tinnitus; tinnitus handicap inventory; tinnitus functional index





## 1. Introduction

Tinnitus is the sensation of a sound in the absence of sound sources internal or external to the body [1] that is prevalent in about 10% of the adult population [2]. This hearing disorder is associated with numerous audiological, cognitive, and neurological problems, ranging from sleeping, listening difficulties, and poor concentration, to stress, anxiety, and depression [3–5].

Although many treatments have been developed to alleviate the symptoms of tinnitus, including sound [6], pharmacological [7], or psychological [8] therapies, there is still no universally effective treatment that can radically cure tinnitus [9]. Sound therapies are currently one of the treatment options used in hospitals and clinics to alleviate tinnitus symptoms. A therapy has been recently proposed, named Enriched Acoustic Environment (EAE) [10,11], that reduces tinnitus-related distress in a high percentage of subjects who complete the treatment [12,13].

There are also neither biomarkers nor objective measures for tinnitus [9]. Therefore, many self-report questionnaires have been proposed to assess the severity of tinnitus, which can be a source of disability on a personal and social level [14]. The most widely used questionnaires are the Tinnitus Handicap Inventory (THI) [15,16] and the Tinnitus Functional Index (TFI) [17,18]. Both have been validated in several languages and recommended for use in clinical and research studies [19].

The THI has 25 items and contains three subscales: functional, emotional, and catastrophic. However, the use of the subscales has been questioned and it is recommended to use the global score [20]. It was developed specifically to diagnose the severity of tinnitus [15,21], but it has been argued to be less sensitive to changes during treatment [17,18]. The TFI, on the other hand, enables the assessment of the functional effects of tinnitus and was developed specifically to detect changes during treatment [22]. It contains 25 items with eight subscales (factors): intrusiveness, sense of control, cognition, sleep, auditory, relaxation, quality of life, and emotional distress. While the final scores of both questionnaires range from 0 to 100, the TFI has higher resolution, as it uses Likert-type responses scaled between 1 and 10 [18].

Thus, while the THI is the most widely used questionnaire [23], the TFI has been claimed to evaluate better treatment-related changes [19]. However, recent studies comparing the global score convergence of both questionnaires showed high convergent validity and strong correlation between them [22,24]. Therefore, the main aim of this study was to compare the ability of the global THI and TFI scores in both intake assessment and responsiveness in a cohort of patients treated with EAE therapy for four months.

## 2. Materials and Methods

### 2.1. Subjects

An observational study, aiming to optimize a treatment for tinnitus subjects (see Section 2.4), is currently being carried by the Acoustic Group of the ITEFI (CSIC), the European University of Madrid, and the Getafe University Hospital. The study has been running since 2018 and recruited participants through a call on the website of the ITEFI. It has the approval of the Ethical Committee of the CSIC and all participants gave their informed consent to publish the results with the corresponding safeguard of the General Data Protection Regulation (EU Regulation 2016/679).

This work included 36 participants, recruited from April 2021 to May 2022, to which the questionnaires THI and TFI were simultaneously administered (see Figure 1). Four of these were excluded due to a recent stapedotomy surgery (1 patient) and for having a THI $\leq$ 20 (3 patients). Ten of the thirty-two remaining participants did not respond to the follow-up request so they were considered as subjects who quit the treatment. Of the 22 participants completing the treatment, 11 were men (mean age = 56.7 years, standard deviation SD = 7.9 years) and 11 women (mean age = 55 years, SD = 10.5 years).

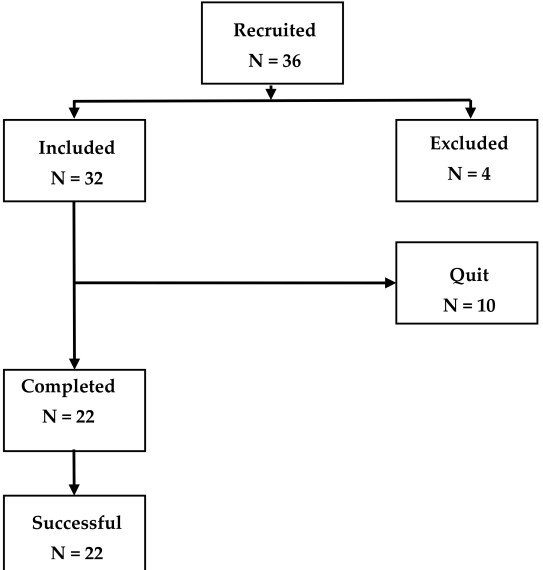

**Figure 1.** Flow diagram of participants in this study.

## 2.2. Audiometric Measurements

All the participants had to provide a recent audiometry, measured by some external audiology or ORL clinic, when they were enrolled. The audiometry consisted of hearing level (HL) values at frequencies (125, 250, 500, 1000, 1500, 2000, 3000, 4000, 6000, 8000 Hz) for both ears. Participants were categorized as either normal hearing (NH) or hearing impaired (HI) depending on their hearing thresholds. The average audiometric thresholds in the whole hearing range ($AAT$), and in the low frequency (from 125 Hz to 2000 Hz) ($AAT_{LF}$) and high frequency (from 3000 Hz to 8000 Hz) ($AAT_{HF}$) ranges were used for this categorization [13,25]. In particular, participants were HI when HL (at any frequency) $\geq$40 dB, $AAT \geq 30$, or $|AAT_{LF} - AAT_{HF}| \geq 17$. Otherwise, the participant was categorized as NH.

Four participants (3 females, 1 male) had NH, while the other 18 subjects (8 female, 10 male) were HI. Their average HL curves are shown in Figures 2 and 3, respectively. Shadowed areas around the HL values display the standard error of mean (SEM) at each frequency (SEM = SD/sqrt(N)). Participants in the HI group exhibited mainly high frequency hearing losses.

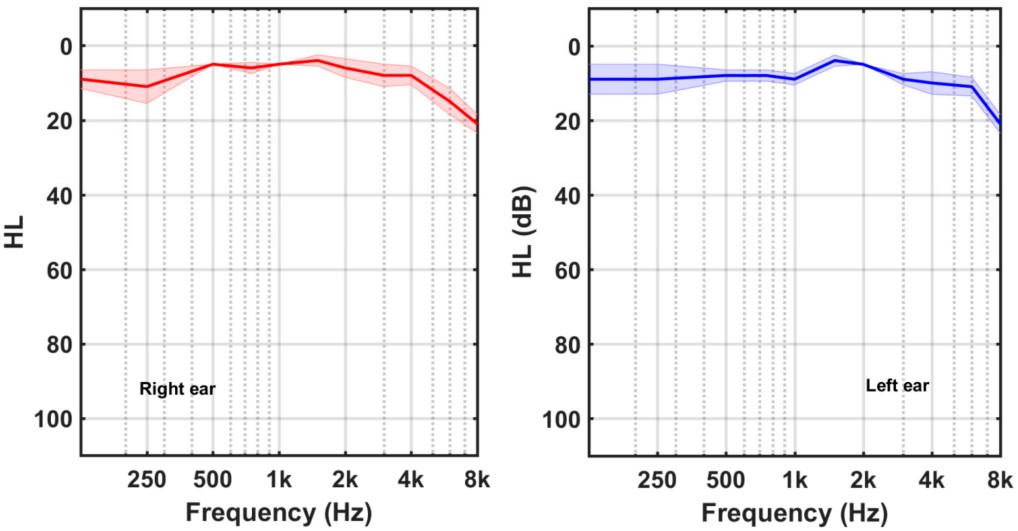

**Figure 2.** Average HL curves of the participants with normal hearing (NH).

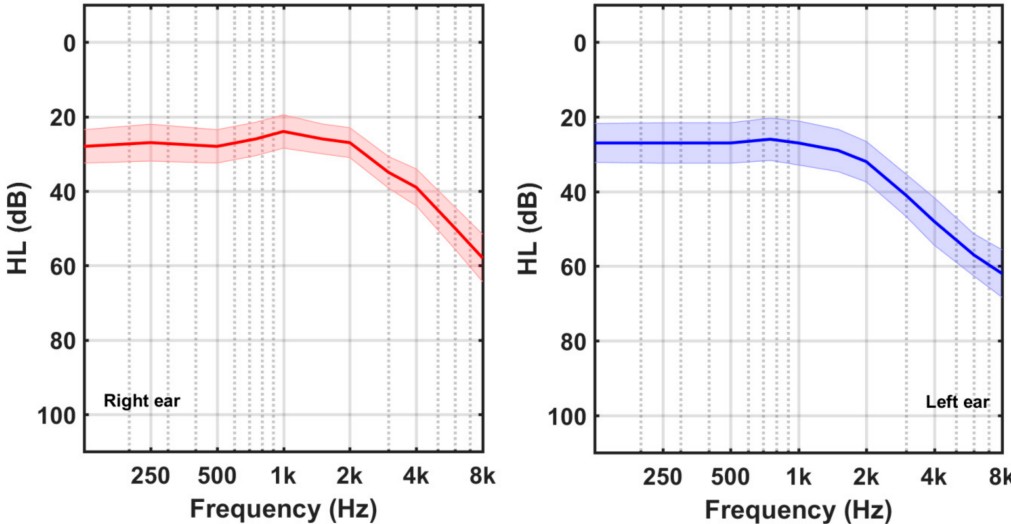

**Figure 3.** Average HL curves of the hearing impaired (HI) participants.

### 2.3. Tinnitus Assessment

Participants were interviewed after enrolling the study about their tinnitus characteristics, including duration, type of sound it sounded like (tonal, ringing, or hissing), its location (left ear, right ear, or bilateral), its possible aetiology, and its baseline severity (using the THI and TFI questionnaires). The information of duration, location, and possible aetiology of tinnitus was directly provided by the participants during the interview. For the tinnitus pitch, custom-designed software was used able to generate tonal, ringing, or hissing sounds [13,25]. In particular, the software generated the sound resulting from passing a random noise through a bandpass filter. The centre and bandwidth of the filter determined the sound type, ranging from tonal (very narrow bandwidth) to ringing (bandpass from 1% to 10% of the frequency centre) and hissing (bandwidth larger than 10% of the frequency centre). The sound type was obtained by matching the sound generated by this software to the own tinnitus sound of the subject.

Table 1 summarizes the tinnitus characteristics of participants. Their mean tinnitus duration was 83 months (SD = 108 months). Their mean tinnitus frequency was 5680 Hz (SD = 2810 Hz). Figures in brackets specify the number of patients having those characteristics. Notice that the total number with a possible aetiology exceeded the total number of participants (22), as some of them said more than one possible tinnitus onset.

**Table 1.** Tinnitus characteristics of participants.

| Duration (Months) | Sound | Location | Possible Aetiology |
|---|---|---|---|
| Mean = 83 SD = 108 | Hissing (9) Ringing (8) Tonal (5) Mean = 5680 Hz SD = 2810 Hz | Bilateral (10) Left ear (8) Right ear (4) | Emotional disorders (9) Hearing loss (7) Conductive troubles (7) Teeth extraction (2) Noise trauma (1) |

The 25-item self-report THI and TFI questionnaires were used for assessing the baseline tinnitus severity of participants at the beginning of treatment. For the THI, a validated Spanish version was provided [26]. For the THI, a validated Mexican version, licensed by the Oregon Health & Science University (OHSU) (TFI NE CR 2020 License), was applied.

Subscales of both questionnaires are not directly comparable, as they are distinct in number and names. However, the final score given by the THI and the TFI can be directly compared, as both have the same number of items and range between 0 and 100, although the name and range of categories are also different (see Table 2).

**Table 2.** Categories of tinnitus.

| THI [21] | | TFI [18] | |
|---|---|---|---|
| Category | THI Range | TFI Range | Category |
| Slight | 0–16 | 0–17 | Not a problem |
| Mild | 18–37 | 18–31 | Small problem |
| Moderate | 38–56 | 32–53 | Moderate problem |
| Severe | 58–76 | 57–72 | Big problem |
| Catastrophic | 78–100 | 73–100 | Very big problem |

Participants were also asked to complete the THI and TFI questionnaires at months 1, 2, 3, and 4 (final) of treatment. The minimum change score for clinical relevance was previously found to be −20 for the THI [27] and −13 [18] or −14 [28] for the TFI.

### 2.4. Tinnitus Treatment

Following the tinnitus neurophysiological model of Jastreboff [27], subjects were treated with a combination of counselling and sound therapy. The main goal of counselling was to reclassify tinnitus into the stimulus-neutral category. If patients could understand

the underlying principles and mechanisms that trigger their tinnitus, they could become desensitized and habituated. In this study, counselling was provided in a single videoconference session of roughly 60 min, just after enrolling. The neural connections between the auditory and limbic systems were explained to the patient, which helped to clarify the connections between emotional aspects of tinnitus and to emphasize the importance of their collaboration.

The main goal of sound therapy was to decrease the intensity of tinnitus-related neural activity. It consisted of an Enriched Acoustic Environment (EAE) stimulus, which could be sequential or continuous [29]. The continuous EAE was a random sound filtered by the HL curves (audiogram) of the patient. According to the computational model of Schaette and Kempter, this stimulus should optimally revert the tinnitus signal [30]. The sequential EAE was comprised of gamma tones with random frequencies within the hearing range and amplitude weighted for the hearing loss values at these frequencies [11,29].

Therefore, both stimuli were customized to the hearing loss of the participant, which is considered to be the main cause of tinnitus. The EAE stimulus was equalized at frequency for each ear and for one ear with respect to the other, so that it could be played in stereo, preferably with good quality circumaural earphones. It was delivered in mp3 format and sent to the participant who listened to it for one hour per day for four months. Each month, the participant sent the completed THI and TFI questionnaires to the researchers to follow up changes in tinnitus perception.

## 3. Results

Figure 4 shows the scatter plot (Figure 4a) and the empirical cumulative distribution function (Figure 4b) of the baseline THI and TFI scores. The correlation coefficient between both scores was r = 0.74 ($p < 0.0001$). Pairs of ($THI_i$, $TFI_i$) scores were closer to the straight line (TFI = THI) at lower and higher values than at middle values. Furthermore, the THI score had more cumulative frequency than the TFI score at medium–low severity grades (severity grade < 60) and vice versa, for high severity grade, the cumulative frequency of the TFI exceeded that of the THI (Figure 4b).

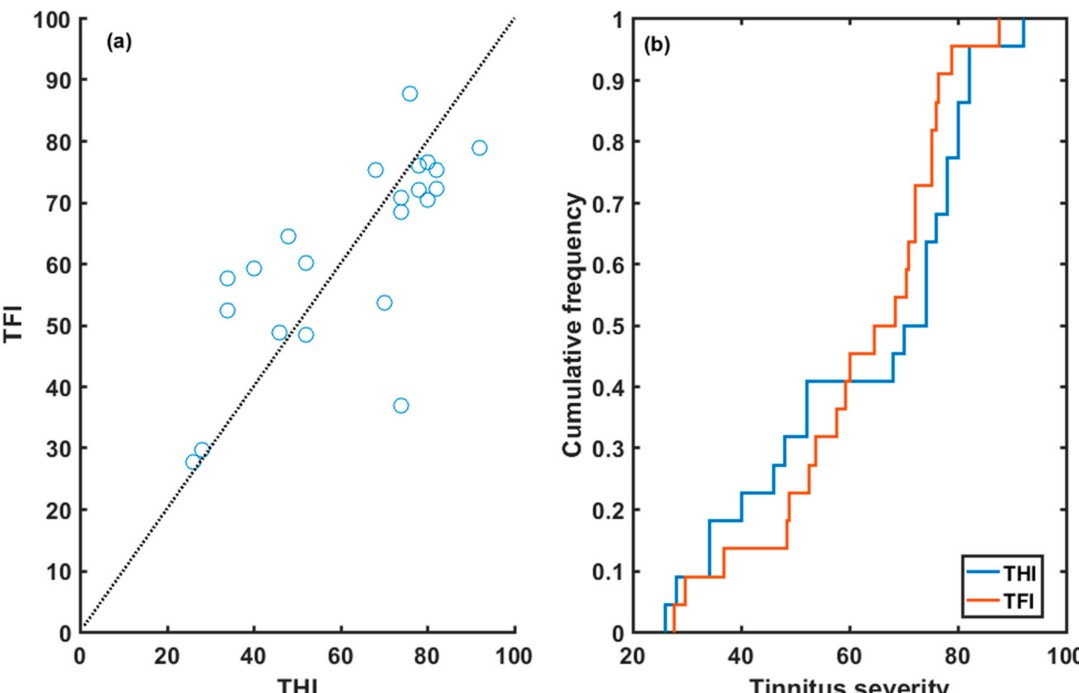

**Figure 4.** (**a**) Scatter plot and (**b**) empirical cumulative distribution functions of baseline THI and TFI scores.

Figure 5a shows the follow-up of the mean THI and TFI scores following the four-month treatment. Shadowed areas around the THI and TFI values display the standard error of the mean (SEM). Both mean scores were similar at baseline (mean(THI$_{initial}$) = 62, mean(TFI$_{initial}$ = 61.9)). After four months of treatment with EAE sound, mean scores were mean(THI$_{final}$) = 32 and mean(TFI$_{final}$) = 36.3. The severity reductions provided by the THI and the TFI scores were 30 and 25.6 points, respectively. Therefore, the EAE therapy provided a clinically relevant reduction in the average tinnitus severity as measured by both questionnaires.

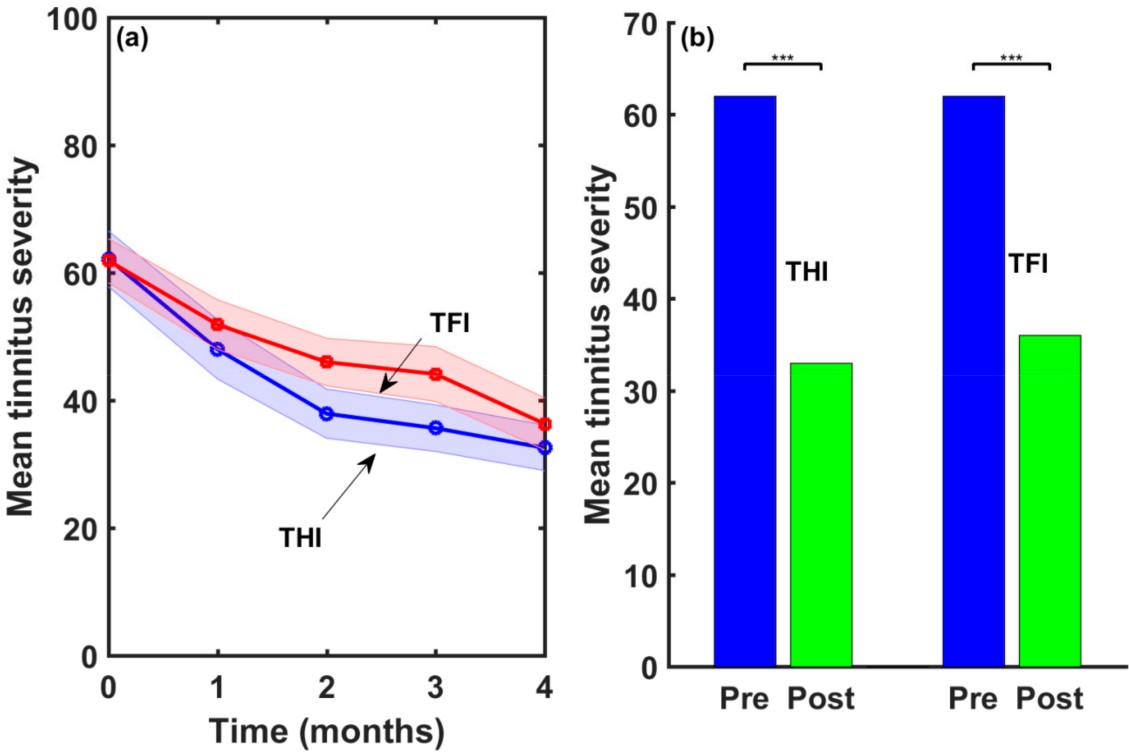

**Figure 5.** (**a**) Tinnitus severity reduction following the four-month treatment and (**b**) result of the application of the ANOVA test.

However, a treatment has to be both clinically relevant as well as statistically significant [31]. Figure 5b shows the result of applying the ANOVA test to the THI and TFI series of scores at baseline (Pre) and after four months of EAE therapy (Post). As can be seen, the THI and TFI reductions provided by the EAE treatment were statistically significant at $p < 0.001$.

Figure 6 displays the fitted straight lines between both questionnaires at baseline (blue line) and after (green line) the four-month EAE treatment. The line TFI = THI (dotted line) is also shown for comparison. Table 3 summarizes the fitting parameters (r = correlation factor, m = slope, b = offset) of both fittings. At baseline, good correlation was found between both questionnaires (r = 0.74). The fitted line to the data crosses the TFI = THI line at THI = TFI = 60. As discussed above, TFI < THI for scores lower than 60, while TFI > THI for scores greater than 60. Therefore, the TFI provided a higher score than the THI for lower severity, while the THI afforded a higher score for higher severity.

Both fitted lines approached the TFI = THI line (dotted line). The correlation coefficient was slightly greater (r = 0.78 versus 0.74) after EAE treatment. Again, the TFI provided slightly higher scores for low severity (TFI < 65), and slightly lower scores for high severity (TFI > 65).

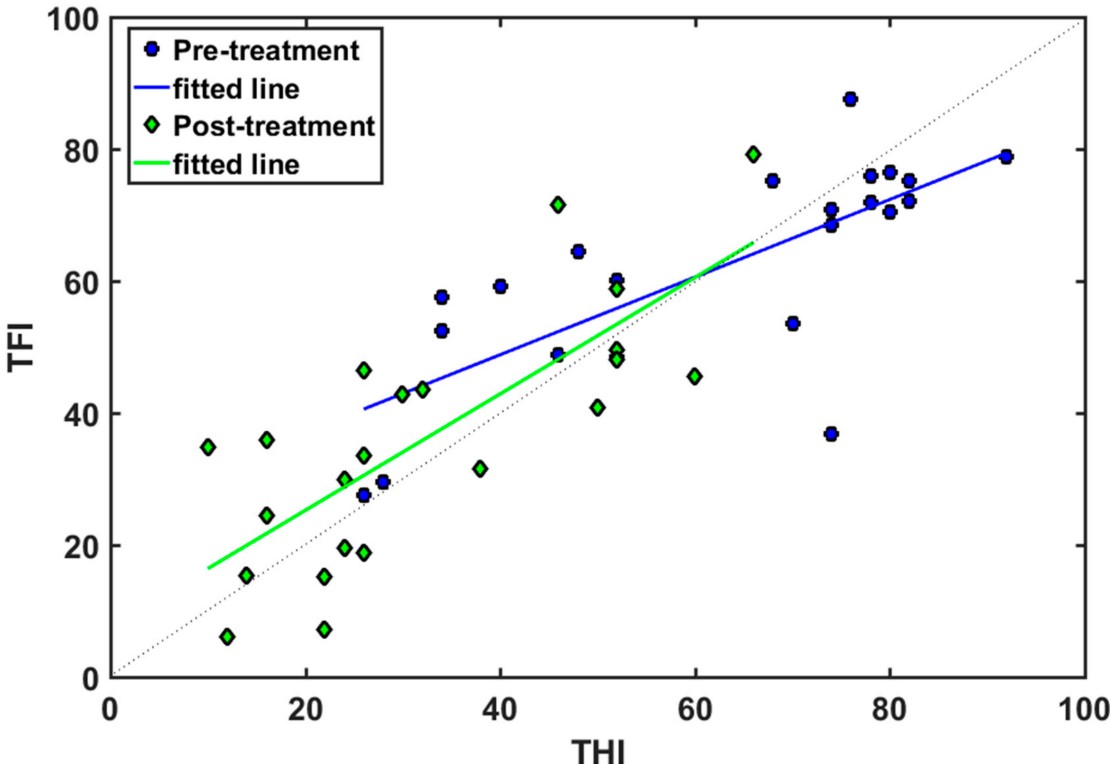

**Figure 6.** (THI, TFI) pair values at baseline (circles, blue) and after EAE treatment (diamonds, green) and their corresponding fitted lines.

**Table 3.** Linear regression parameters of TFI versus THI at pre- and post-treatment.

|  | r | m | b | Regression Line |
|---|---|---|---|---|
| Pre-EAE | 0.74 | 0.59 | 25 | TFI = 0.59 THI + 25 |
| Post-EAE | 0.78 | 0.88 | 7.6 | TFI = 0.88 THI + 7.6 |

## 4. Discussion

This is a non-controlled observational study with a small number of participants. Nevertheless, this is part of an ongoing study on the efficacy of the EAE as a sound treatment for tinnitus, which has used THI since the beginning as a tinnitus outcome. The TFI was incorporated recently (April 2021) once the license was obtained from the OHSU, so the number of patients to which both questionnaires were supplied is low. The sample size will increase as the last recruited patients complete their four-month treatment.

In previous studies [13,25], patients were categorized in NH and HI groups. In these studies, 71–73% of subjects were included in the HI group, and 27–29% belonged to the NH group, in accordance with figures provided by other authors [9,32]. In this sample, 18.2% of patients had normal hearing (NH) and 81.8% were hearing impaired (HI). Nevertheless, it is necessary to emphasize that normal hearing at the discrete frequencies mentioned in Section 2.2 does not exclude, (1) hearing losses at frequencies above 8 kHz [33], (2) hidden hearing losses [34], or (3) missed hearing losses at other frequencies than measured in the audiogram [35], which could also trigger tinnitus.

The application of EAE to this sample confirms the good results in previous studies. In particular, the application of EAE therapy in 25 patients, one hour per day for four months, improved the tinnitus distress related in $\Delta THI = THI_{pre-EAE} - THI_{post-EAE} = -29$ points [12]. Similarly, when EAE was applied to a cohort of 83 patients, with the same listening time, $\Delta THI = -23$ [13]. In this study, a higher reduction is obtained in both THI and TFI as a result of the EAE therapy. In particular, $\Delta THI = -30$ and $\Delta TFI = -25.6$ points.

In accordance with other studies, a good correlation was obtained between the THI and TFI questionnaires at baseline and after the treatment [22,24]. At baseline, the THI provided a higher score than the TFI for a higher tinnitus grade but a lower score for lower tinnitus severity. All the post-treatment scores were lower than the pre-treatment scores. According to these results, both the THI and the TFI were good questionnaires for intake assessment as well as for treatment-related changes (see Figure 5). The THI provided a slightly higher score drop than the TFI after the treatment. However, the TFI had better resolution, as the responses of the THI to each item take values (0,2,4), while the responses to the TFI vary between 1 and 10.

## 5. Conclusions

The current study compared the ability of THI and TFI global scores in both the assessment of intake and responsiveness in a cohort of patients treated with EAE therapy over four months. The results confirmed that both questionnaires displayed a good correlation (r = 0.74) and provided almost identical results as evaluators of the tinnitus severity at baseline. Moreover, both questionnaires were able to detect changes related to the application of the EAE treatment, with a slightly larger severity decrease in the THI ($\Delta$THI = $-30$, $\Delta$TFI = $-25.6$). Both score drops were clinically relevant and statistically significant.

**Author Contributions:** Conceptualization, P.C. and M.C.; methodology, P.C.; software, P.C. and M.F.; validation, R.S.; formal analysis, P.C. and M.C.; investigation, P.C., M.F., M.C. and R.S.; data curation, R.S. and P.C.; writing—original draft preparation, M.F and P.C.; writing—review and editing, M.C. and R.S.; supervision, P.C. and M.C. All authors have read and agreed to the published version of the manuscript.

**Funding:** M.F. has been funded by the European University of Madrid.

**Institutional Review Board Statement:** The study was conducted according to the guidelines of the Declaration of Helsinki, and approved by the Institutional Ethics Committee of the CSIC (reference 004/2022).

**Informed Consent Statement:** Informed consent was obtained from all subjects involved in the study.

**Data Availability Statement:** The data are not publicly available due to the confidentially clause of the Informed Consent Form.

**Acknowledgments:** Volunteers who participated in this study are kindly acknowledged.

**Conflicts of Interest:** The authors declare no conflict of interest.

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
