# Peer review of "Comparison of Tinnitus Handicap Inventory and Tinnitus Functional Index as Treatment Outcomes"

_audiolres, doi:10.3390/audiolres13010003_

Round 1

Reviewer 1 Report

The manuscript aims to compare the THI and TFI scores of a group of participants who were treated due to their tinnitus with a combination of counselling and sound therapy. 

It is mainly dealing with this comparison between the 2 scales, however, it should be revised before considered for publication due to several factors:

The manuscript should be edited by an English speaking editor, since there are quite a few sentences which are not clear enough, or in which the use of words should be more precise. Also, please keep using the past tense in the Materials and Methods, Results and Discussion sections.

Generally, The authors describe the treatment as sound therpay, however, it includes a counselling session. This should be corrected throughout the manuscript.

Abstract:

The abbreviations EAE, THI and TFI are not explained. Line 18 in the abstract: "four" and not "fourth month" The conclusion part of the abstract is not clear, and I suggest refraining from the use of the word "seems".

Introduction:

Line 37 - it is not the main treatment, but rather one of the treatment options.

Line 39 - what does high mean? Again, refrain from "seems" as it is too unclear to the reader.

Line 40 - comlpleting and not finishing

Line 50 - complained for being - change to: argued that it is, and please add references to backup this statement

Line 51 - is also able - change to enables

Line 51-52 - with instead of disributed into

Line 55 - higher instead of more

Line 57 - "a secondary" 

Line 58 - to discriminate between changes

Line 58-59 - what do you mean by more efficiently? Could you add a reference to that?

Line 63 - for and not along

Materials and methods

Line 66 - Again - is it only a sound therapy?

Line 69 - through and not trough

Line 75 - add "patient" after 1 and "patients" after 3

Line 76 - Please rephrase the sentence.

Line 78 - the brackets around SD are not needed

Figure 1 is not clear and not accurate since it can be understoood that out of 4 patients who were excluded 10 more quit the traetment. Also please use "completed" instead of "finished".

Line 84 - HL - is this hearing level or hearing loss?

Lines 89-90 - Why were those specific thresholds were chosen?

Line 91 - females and males.

Line 102 - "sounded" and not "looked"

Line 103 - How was the baseline severity was measured? I am sure it was using the THI and TFI, but it should be mentioned.

Table 1 - the word "trouble" should be changed

Also, summing up the sounds gives 23, but there were 22 participants.

Line 121 - TFI and not THI

For those of us not speaking Spanish - is there any meaning to the fact that the Mexican version was used in Spain? Please explain in the Discussion section.

Line 129 - At months 1, 2, 3, 4 (and not 1st, 2nd etc.)

Line 130 - I suggest writing: "was previously found to be" instead of "stated in" 

Line 146 - "was comprised of"

Line 148 - Is the abbreviation HL here referring to hearing level or hearing loss?

Also - what about patients who do not have hearing loss? How were they treated?

Results:

Generally, the results are described in too many details. The reader is supposed to be referred to the figures and not to read their description in the text. Please rephrase the paragraph.

Line 156 - Figure 4 and not 4a

Line 161 - than and not then

Line 173 - This sentence is usually not a part of the Results paragraph.

Line 174 -175 - series of scores, and not participants

Figure 5 - Y axis - Mean tinnitus severity

Line 181 - Comparing rather than: It can be interesting to

Line 187 - lower and not lesser

Line 195 - TFI=THI

Discussion:

Line 219-220 - This is not the aim of the study and thereofre the discussion should not be about that. In addition to that , the sentence is grammatically incorrect.

Generally, the Discussion repeats the description of the results, and therefore should be changed to discuss the reasons for the findings, focusing on the THI and TFI. Please discuss how those findings can be explained (apart from the item's scoring).

Conclusions:

The first paragrpah is too long and should be shortened.

The results are once more described, instead of concluding in a concise paragraph. 

Author Response

Please, see the attached document

Reviewer 2 Report

In this research paper, THI and TFI scores before and after a form of sound therapy for tinnitus are compared. The authors conclude that both questionnaires are appropriate for assessing baseline tinnitus severity and treatment-related changes, and scores from both questionnaires are highly correlated. This is in line with earlier reports comparing these questionnaires.

My main concern with this research paper is that the primary aim of the paper seems poorly defined. The authors seem to place the emphasis on comparing THI and TFI values. Nevertheless, at L166-177, they describe the effects that their sound treatment had on tinnitus severity in some detail, and they conclude that this therapy has both clinically and statistically significant effects. But at several points throughout the manuscript, they explain that this study is still ongoing. Why did the authors decide to publish these data already, if the study is still running? Are they planning on re-publishing these data after completion of the study? (At L204-205, they write “The sample size will increase as the last recruited patients complete their four-month treatment.) This would be highly questionable!

I would urge the authors to better explain why they chose to publish these data at this point in time, when the study is still running. If there is a (published) protocol of the study available, I would also ask them to provide a reference to this protocol and substantiate whether it allows the publication of this 'interim' analysis. If they want to go through with the publication of these data at this point in time, I would strongly advise them to consider the implications for the results of this study once it is concluded (i.e., it would be highly questionable to re-publish results based on the same data). Please provide some comments on this issue.

I provide further, more minor comments below:

Abstract

L19-20: “30 and 2.6 point reduction in THI and TFI, respectively”: According to the results, this should be 30 and 26 point reduction, please adjust

Introduction

L36-37: “Sound therapies are currently the main treatment used in hospitals and clinics”: This may be too strong of a statement. Main treatments might vary from region to region and in my experience, psychological treatments might be more commonly prescribed. Please consider adjusting this statement

Methods

Figure 1: Please adjust the flowchart to show drop-outs at the adequate place (i.e. not after exclusion but rather after inclusion). Moreover, the drop-out rate is reasonably high. This might be a source of bias regarding the results. Could the authors provide the reasons why 10 out of 32 participants dropped out before finishing the study?

Results

Figure 6 and L195-196: “Notably, the fitted line approaches the TFI = THI line after treatment”: To me, the difference in fit is not really noticeable. The difference in correlation coefficients is also not very big (according to Table 3, 0.74 vs. 0.77, although a coefficient of 0.78 is given in the text at L196 – this might be a rounding error, please double check this). If the authors make the claim that the fit is better for the post-treatment scores than for the pre-treatment scores, could they add any statistical calculations comparing both fits so that this putative difference between the two can be proven?

Discussion

L224-225: “After treatment, all scores [...] seem to line up better with the THI = TFI line”: In accordance with my previous comment, this claim does not seem to be substantiated by the results

L227: “The THI seems to provide a higher score drop than the TFI along the treatment.” (also mentioned in the abstract): The difference between the drop in THI scores (30) and TFI scores (26) seems minor to me, and possibly negligible. Could the authors add some statistical investigation into whether there really is a difference between the two?

Author Response

Please, see the attached document

Reviewer 3 Report

It is a very good study, original in the project and design.

The results reported are very interesting but must be considered preliminary since the sample is small. As reported by the same Authors usually in questionnaire studies the sample must be larger in order to obtain more reliable results. My suggestion is to wait in order to publish a paper with a larger sample since the objective of the study is really interesting.

The Authors have chosen a complex methodology for evaluating the normality of hearing function versus the presence of hearing loss. Looking at fig. 1 the normal group presents a mean threshold value under 25 dB at each frequency tested. therefore I suggest, as usually done, to consider as cut-off for normality the presence of a threshold less than 25 dB at each frequency tested. The evaluation of threshold differences at low and high frequencies, as evaluated in this paper, is not considered in the result and in the discussion chapters therefore can be considered as not useful in the study.

Moreover, the Authors have accepted audiometries done in other centers therefore there is not enough reliability about the reliability of these tests. 

Author Response

Please, see the attached document

Round 2

Reviewer 1 Report

The title should be re-phrased

Abstract line 25: better and not more

Introduction: line 37: one and not ones

39: releases - I suggest choosing another word

57: "at least a secondary" - what do the authors mean? Please rephrase

58-59: re-phrase

Materials and Methods:

line 57: I suggest writing: "to optimize a treatment for tinnitus"

78: of and not for

Figure 1: emit the successful box

Results:

line 228: please re-phrase to: "all the post-treatment scores were lower than the pre-treatment scores"

line 231: after and not along. better and not more

Conclusions: Still too long and repeating the results. They shhould be concise.

Reviewer 2 Report

I thank the authors for addressing my comments. I have some remaining questions for the authors below:

(-) In the ongoing study, have I understood correctly that participants are now only filling out the TFI and no longer fill out the THI? That would indeed explain why they choose to publish these data now, as the participants they report on are the only ones who have filled out both questionnaires. If all participants in the ongoing trial are still filling out both questionnaires, I would still consider waiting until trial completion to publish these data.

(-) Thank you for addressing the drop-out rate in your comments. Would it be possible to add data on how many participants did not finish the treatment vs. how many finished the treatment but did not complete the follow-up questionnaires? Or, if it is not possible to distinguish between both groups, could you add that to the manuscript?

Reviewer 3 Report

The Authors have answered to the observations.

Now the paper should be considered for publication.

Author Response

Thank you very much for your comments